# Stigma and Discrimination towards People Living with HIV in the Context of Families, Communities, and Healthcare Settings: A Qualitative Study in Indonesia

**DOI:** 10.3390/ijerph18105424

**Published:** 2021-05-19

**Authors:** Nelsensius Klau Fauk, Karen Hawke, Lillian Mwanri, Paul Russell Ward

**Affiliations:** 1College of Medicine and Public Health, Flinders University, GPO Box 2100, Adelaide, SA 5001, Australia; fauk0001@flinders.edu.au (N.K.F.); lillian.mwanri@flinders.edu.au (L.M.); 2Institute of Resource Governance and Social Change, Jl. R. W. Monginsidi II, No. 2, Kupang 85221, Indonesia; 3Infectious Disease—Aboriginal Health, South Australian Health and Medical Research Institute, Adelaide, SA 5000, Australia; karen.hawke@sahmri.com

**Keywords:** HIV stigma and discrimination, people living with HIV, social process, Indonesia

## Abstract

HIV stigma and discrimination are a major challenge facing people living with HIV (PLHIV) globally. As part of a larger qualitative study with PLHIV in Yogyakarta and Belu, Indonesia, this paper describes the participants’ perceptions about drivers of HIV stigma and discrimination towards them within families, communities and healthcare settings, and highlights issues of HIV stigma as a social process. Participants were recruited using a snowball sampling technique. Data analysis was guided by the framework analysis for qualitative data, and conceptualization and discussion of the study findings were guided by the HIV stigma framework. The findings showed that participants experienced stigma and discrimination across settings, including in families and communities by family and community members, and in healthcare settings by healthcare professionals. The lack of knowledge about HIV, fear of contracting HIV, social and moral perceptions about HIV and PLHIV were perceived facilitators or drivers of stigma and discrimination towards PLHIV. HIV stigma and discrimination were also identified as a process linked to the whole groups of people within families or communities, which occurred within social context in Yogyakarta and Belu. The findings indicate the need for HIV education for family and community members, and healthcare providers to enhance their knowledge of HIV and improve acceptance of PLHIV within families, communities and healthcare settings.

## 1. Introduction

Human immunodeficiency virus and acquired immune deficiency syndrome (HIV/AIDS) have been reported as a major public health problem for decades, with an estimated 38 million people globally living with the infection [1,2]. It is also well acknowledged that a diagnosis of HIV infection causes a range of detrimental impacts on people living with HIV (PLHIV) [3,4,5,6]. Stigma which is often manifested as discrimination or unfair treatment by other (HIV-negative) people is one of the major negative impacts on PLHIV in many settings [3,7,8]. Despite positive achievements in the response to the epidemic globally, increased universal access to antiretroviral therapy (ART) and reduction of infection across the world, HIV stigma and discrimination are still a global problem [2,9].

Previous studies and reports have suggested that HIV stigma and discrimination towards PLHIV occur within families, communities, and in healthcare settings [10,11,12]. Several studies have reported that stigma and discrimination towards PLHIV often occur within families by parents, siblings, relatives or in-laws [13,14]. These are reflected in a range of discriminatory treatment and behaviors, including refusal by others to share food and rooms with PLHIV, separation of personal belongings and eating utensils of PLHIV from those of other family members, isolation of PLHIV by their own family, including exclusion from usual family activities such as cooking and family gathering [13,14,15,16,17,18,19,20]. HIV stigma and discrimination towards PLHIV have also been reported to be inflicted by neighbors, friends, and co-workers and these often manifest as rejection, neglect, avoidance, ridicule, verbal abuse, insult and harassment [14,15,16,17,19,21,22]. Similar acts of stigma and discrimination towards PLHIV have also reported within healthcare facilities or settings by healthcare professionals in a variety of ways, including criticizing, blaming, shouting at or throwing health records on patients’ faces, and neglecting or refusal of care and treatment, and through unnecessary referral to other healthcare facilities [14,18,21,23,24]. The fear of contracting HIV through physical, social and healthcare-related contacts and interactions, and the lack of knowledge about how HIV is transmitted, have been reported as the main drivers of HIV stigma and discrimination in these settings [13,14,25,26,27]. HIV stigma and discrimination have also been reported to cause negative impacts on psychological state, health outcomes, and social life of PLHIV. They are associated with stress, anxiety, depression and low quality of life for PLHIV [3,4,5,6]. They have been reported to negatively influence access and adherence to ART or HIV prevention and treatment efforts and disrupt social relationships of PLHIV with their families, relatives, friends and neighbors [3,4,5,6].

Although many other studies have explored HIV stigma and discrimination towards PLHIV in different settings, most of these studies have focused on stigma at individual level, including studying the attitudes and behaviors of HIV non-infected individuals towards PLHIV [8,11,28], leaving the gap in knowledge about how HIV stigma and discrimination are enacted as a social process. Therefore, the aim of this study was to explore HIV stigma and discrimination beyond individuals and to assess how they are enacted as a social process in the context of families, communities, and healthcare settings. The overall aim is to contribute to the understanding of drivers of stigma that arise within social contexts in Indonesia where the influence of strong family and community values, norms, ties, and religious thoughts on stigma and discrimination towards PLHIV have not been addressed in previous studies [8,11,28]. As HIV stigma and discrimination are reported to occur across settings in Indonesia [14,27,29,30,31] and Indonesian society is influenced strongly by family and community values, norms, ties, influences, and religious [32,33,34], it is important to unpack this complex societal structure, to further inform how social processes influence and propagate discriminatory and stigmatizing attitudes and behaviors towards PLHIV. This information is crucial as will provide significant contribution to the current body of knowledge on the topic and inform policies and practices within government and non-governmental institutions and organizations to address social impacts of HIV and improve health outcomes of PLHIV in Indonesia and globally.

## 2. Methods

The report of the methods section was guided by consolidated criteria for reporting qualitative studies (COREQ) checklist [35]. The checklist contains 32 items (Appendix A) that need to be covered to support the explicit and comprehensive reporting of qualitative studies [35].

### 2.1. Conceptual Framework

The HIV stigma framework developed by Earnshaw and Chaudoir [3] was used to guide data collection, conceptualize, and discuss the findings of the current study. This framework suggests that stigma is a devalued attribute which had detrimental effects on PLHIV through various stigma mechanisms [3,27,36]. Stigma mechanisms reflect a psychological response of HIV-negative people towards PLHIV or the devalued attribute, and to the possibility of HIV transmission by PLHIV [27,37]. Such responses are often manifested as prejudice, stereotyping and discrimination towards PLHIV [3,7,27]. Prejudice refers to negative emotions or feelings of uninfected people, such as disgust, anger, and fear towards PLHIV [38]. Stereotype is the negative beliefs (e.g., PLHIV have deviant behaviors or are dangerous) which are often attributed to PLHIV [39]. Discrimination refers to unfair treatment of uninfected people towards PLHIV or the belief of PLHIV about their experience of prejudice and stereotype by HIV-negative people [3,7,8]. This framework suggests that discrimination or unfair treatment, also known as external or enacted stigma, is a mechanism through which PLHIV experience HIV stigma [3,7,8]. It is also a process of influence of social norms and values, and institutional policies on negative attitudes and behaviors of others (e.g., family members, community members and healthcare professionals) towards PLHIV, which negatively impacts behaviors, psychological health outcomes and social life of PLHIV, and constrains their opportunities (e.g., access to healthcare) and wellbeing [40,41].

### 2.2. Study Design, Recruitment of the Participants, and Data Collection

This paper presents data from a large-scale qualitative study exploring the views or perceptions of PLHIV about HIV risk factors and impacts and their access to HIV healthcare services in Yogyakarta and Belu, Indonesia. The qualitative design was used as it has been found appropriate and effective when exploring participants’ perspectives and deep insight of their real-life experiences [42,43]. It facilitated the exploration of the participants’ stories, understandings and interpretations about the supporting factors for HIV transmission among them, and drivers of stigma and discrimination against them by other (non-infected) people [44,45,46]. It also enabled the researchers to explore and understand values and meanings the participants had in relation to HIV stigma and discrimination facing them in their daily lives [42,47].

PLHIV who participated in this study were recruited using the snowball sampling technique. The recruitment process started after a permission letter was solicited from HIV clinics in both study settings. The study information packs were distributed to potential participants who accessed HIV healthcare services in the clinics through the receptionist at each HIV clinic. Potential participants who called and confirmed their willingness to participate in the study were asked to recommend a preferred time and place for an interview. The recruitment process took three months, with 92 PLHIV (52 women and 40 men; 46 in Yogyakarta and 46 in Belu) participating in the study.

Data collection was conducted from June to November 2019 using one-on-one in-depth interviews in a rented house close to the HIV clinic in Yogyakarta and a private room at the HIV clinic in Belu. The interviews were conducted by the first author (NKF) and in Bahasa (the primary language of the researcher, who also speaks fluent English) and audio recorded digitally, and notes were taken during the interviews. The duration of the interviews ranged from 35 to 87 min. Regarding HIV stigma and discrimination, the interview topics explored participants’ perceptions and experiences of HIV stigma and discrimination. The researcher probed further about attitudes and behaviors of family and community members and healthcare providers towards them. Moreover, the impacts of unfair treatments and attitudes of other people towards them were explored. Additionally, participants were asked about perspectives regarding drivers or facilitators of and mechanisms or processes through which those facilitators or drivers contributed to stigma and discrimination. Participants’ perspectives about how social influence among family and community members which led to stigma and discrimination against them and other PLHIV were also explored. The development of interview questions was informed by the findings of previous studies and the theoretical framework used in this study. Recruitment of the participants and interviews ceased once the research team felt that the collected data were rich enough and data saturation had been reached. This was reflected in the similarities of information provided by the participants in the last few interviews, which justified our decision to cease data collection at that point. No repeated interview was conducted with any of the participants. Two potential participants withdrew their participation due to personal reasons. We did not offer an opportunity for participants to read and correct the information provided after the transcription due to the sensitivity of the topic and to prevent the possibility of the transcripts being received and read by their family members, which might divulge the participants’ HIV status, in case they had not disclosed it to family members. There was no established relationship between the researcher and any of the participants prior to the study.

### 2.3. Data Analysis

The digital recordings of the interviews were manually transcribed verbatim in Bahasa by the first author (NKF). The transcripts were then imported to NVivo 12 where the comprehensive data analysis was performed, which was guided by a framework analysis for qualitative data by Ritchie and Spencer [48]. The framework was used as it helped the management of qualitative data in a coherent and structured way, and guided the analytic process in a rigorous, transparent, and valid way. This framework suggests five steps of qualitative data analysis, which are: (i) Familiarization with the data or transcripts, which was done by repeatedly reading each transcript, breaking down the data into small chunks of data, and making comments or labels to the data extracts of each individual transcript. The transcription of the audio recordings, which was manually performed using a laptop, had been started by the field researcher following each interview during the data collection process. At this stage, emotions or notes undertaken during interviews were added into each individual transcript. Thus, the process of familiarization with the data had been started alongside the data collection process; (ii) identification of a thematic framework by writing down key issues and concepts that recurrently emerged from the data, which was performed after importing each individual transcript with the initial comments, codes, labels into NVivo. Themes that were used to form the thematic framework were derived from both the HIV stigma framework used in this study and the collected data. The identification of the thematic framework was an iterative process that involved changing and refining themes; (iii) indexing the data which was comprehensively performed using NVivo. The process of indexing (coding) was started by making open codes to data extracts of each individual transcript resulting in a long list of open codes or nodes. This was followed by close coding to identify similar or redundant nodes or codes and reduce the long list of open codes to a manageable number, and then, nodes or codes that seemed to fall into the same themes and sub-themes were grouped together; (iv) charting data through arranging appropriate thematic references in a summary of chart which enabled comparison across interviews and within each interview. This was performed by reorganizing and summarizing codes from each individual interview transcript, which had been grouped into separate themes in the previous section, and putting them together under each theme; and (v) mapping and interpretation of the data through which data were examined and interpreted [48,49]. Based on the pieces of data that had been indexed and charted in the previous steps, the researchers systematically pulled together key characteristics of the data, mapped, and interpreted data set as a whole. Transcription, coding and analysis were conducted in Bahasa, and quotations for publication purposes were translated into English by NKF. To maintain the accuracy of the translation and credibility of the findings, checking and rechecking transcripts against the translated interpretations or examination of meaning in both source (Bahasa) and target (English) languages were done during the analysis [50]. Analysis was primarily undertaken by NKF, although team-based analysis was undertaken at regular research team meetings whereby all authors undertook independent analysis and then team decisions were made about the validity of the final themes and interpretation.

### 2.4. Ethical Consideration

The participants were informed that the study obtained the ethics approvals from Social and Behavioral Research Ethics Committee, Flinders University (No. 8286), and the Health Research Ethics Committee, Duta Wacana Christian University (No. 1005/C.16/FK/2019). They were advised about the purpose of the study and the voluntary nature of their participation prior to the interviews through the study information packs distributed to during the recruitment process and again in person by the field researcher prior to each interview. They were informed about their right to withdraw from the study at any time, without consequence, if they felt uncomfortable with the questions being asked during the interviews. They were also advised about the approximate interview duration of 45 to 90 min and that audio recording and notes would be undertaken during the interviews. The participants were assured about confidentiality and anonymity of the data or information they provided in the interview, which was maintained by assigning a specific study identification letter and number for each participant (e.g., FP1, FP2, …., or MP1, MP2, …. FP stands for Female Participant and MP stands for Male Participant). Reimbursement of IDR 100,000 (±USD 7) for transport and time of the participants was provided for each of them after the interview. The informed consent form was provided for each participant, signed, and returned to the researcher prior to the interview.

## 3. Results

### 3.1. Demographic Profile of the Participants

A total of 52 women and 40 men living with HIV aged between 18 to 60 years old were interviewed. Most of the participants’ age was between 30 and 49 years (68 people) and over a half of them were married or remarried (48 people), while the rest were non-married (divorced, widowed or single). Most of the participants had been living with HIV between one to five years (55 people), and a number had been diagnosed with HIV for six to 10 years (25 people) and 11 to 15 years (12 people). Several reported being also diagnosed with other sexually transmitted infections: herpes, candidiasis, syphilis and gonorrhea (13 people). Twenty-eight people had also been infected with tuberculosis. All the participants reported being on ART.

Education backgrounds varied, with 37 of the participants graduating from senior high school, 21 from university, 18 from junior high school and the rest graduating from elementary school or dropping out of school. Twenty-one women reported being housewives, and the others engaged in different kinds of professions: entrepreneur, tailor, NGO worker, sex worker, health worker, shop keeper, private employee, banker, civil servant, and laundress. Eleven men reported being entrepreneurs, nine were drivers (taxi, truck or motorbike taxi) and the rest engaged in a variety of professions as teacher, farmer, police, private employee, and iron welder. Most participants interviewed in Yogyakarta and Belu were Muslim and Christians, respectively.

### 3.2. Stigma and Discrimination within Families

Participants across both study settings acknowledged the existence of stigma and discrimination towards PLHIV, and some had experienced stigma or discrimination within families or communities or healthcare facilities. For example, relatively similar numbers of both women (*n* = 17) and men (*n* = 12) interviewed in Yogyakarta (*n* = 13) and Belu (*n* = 16) reported having experienced the same HIV stigma and discrimination within families by their parents, siblings, and in-laws at a certain point of time following the HIV diagnosis. Separation of personal belongings such as clothes and eating utensils from those of other family members, separation from children, ostracism, avoidance, negative labelling as sex worker, and being asked to stay away from home or live in other places, were some instances of discriminatory and stigmatizing attitudes and behaviors of family members towards them:

*“I was separated from my child (by her sister-in-law). My child slept with her aunty. My eating utensils were given a sign. The relatives of my husband also said to my sisters-in-law: ‘the spoon she used should be separated, you can be infected’. They were nice in front me but felt disgusted about me at the back. They asked my sisters-in-law to chase me and my husband (her husband was HIV-negative) away from the house (the woman and her husband lived together with her sisters-in-law in the same house)”*.(FP5, married, Yogyakarta)

*“I experienced it (discrimination), it felt painful here (pointing to his chest). I was ostracized by my family: my father, mother, brothers and sisters. At that time, I did not have the spirit to live anymore because I was ostracized like that. It took place for a long period of time till I remarried. …. At the first time I told them that I contracted HIV, everybody was shocked and angry at me. I was nearly chased away from home. Finally, my father started to accept me step by step, but my mother was scared of being avoided by other people if my HIV status is known to them (other people within their community). My oldest and youngest brothers and sisters avoided me. My plate was separated, and I washed it myself. I was scolded and asked not to use the toilet, if I used it then I had to clean it up afterwards (The man and his second wife live separated from his family after they got married)”*.(MP19, remarried, Yogyakarta)

*“My father, mother and nephew know about my (HIV) status. My eating utensils, food and water are separated from those of other family members. They separate all of these, I feel so painful and sad, but I cannot do anything and just accept it. I feel like my family members do not really care about me. They do not care whether I eat or not and this makes me cry sometimes”*.(MP5, single, Belu)

Fear of contracting HIV was reported by the participants as a major facilitator of unfair treatment or discrimination by family members towards them. Such fear was reported to be supported by the lack of knowledge of family members about HIV or the means of HIV transmission. The participants also described that their family members were easily influenced by inaccurate HIV information spread by others within their family or dependent on information heard from neighbors or other community members as source of HIV knowledge, which reflected the lack of knowledge of family members about HIV:

*“We (the woman and her husband who was also HIV-positive and had died from AIDS) were avoided by nearly all the family members of my husband because they were scared of getting HIV, they did not know how it is transmitted. They thought they would get it if they have physical contact with us. A relative of my husband was the one who spread this misleading information to all the family members of my husband, she told all of them this wrong knowledge, hence they were influenced by what she said. Families and neighbors here are very close to each other, so sensitive information like this (about HIV) can quickly spread and they can easily influence each other and believe it”*.(FP4, widowed, Belu)

*“My food was given to me (by her mom) through the bottom of the door, just like you would do for a dog. It was very painful and if I remember this (how her mom treated her), I sometimes still feel the pain. But she is my mom, she knew very little about HIV. …. What she knew about HIV was just based on information she had heard from neighbors and other people around us in the community. Information and perceptions from others influenced her reaction towards me once I told her I have HIV”*.(FP24, divorced, Yogyakarta)

*“I was discriminated by my parents. My father collected all my clothes and boiled them with hot water. He was asked by his second wife to do so. His second wife does not want to accept me. It is apparent that they are scared of me transmitting the virus to them because they do not know that HIV does not transmit through clothes”*.(MP4, married, Yogyakarta)

### 3.3. Stigma and Discrimination within Communities

Several participants in Yogyakarta (*n* = 7) and Belu (*n* = 17) also reported having experienced stigma and discrimination within communities where they lived and interacted. For example, both women (*n* = 13) and men (*n* = 11) described that they were rejected by neighbors, friends, or other community members due to their HIV-positive status, reflected in the acts of refusal of direct physical contact such as shaking hand and eating food they had touched, avoidance or keeping distance, and exclusion from social and community activities. The spread of untrue stories and gossip about their HIV status, were also other instances of discriminatory and stigmatizing attitudes and behaviors of neighbors, friends, and other community members towards them. However, some participants in Yogyakarta who worked for HIV programs also described that although stigma and discrimination towards PLHIV still existed with communities, they seemed to be decreasing due to increased responses (HIV programs) to the issue, which could be the reasons why fewer number of participants in the setting reported the experience of stigma or discrimination within their communities:

*“I got discrimination in the community where I lived before. If I had touched any foods, then people would not eat those foods. Some (community members) spread information that I am HIV-positive and gossiped about it. I experienced these for about two years”*.(FP17, divorced, Yogyakarta)

*“There are neighbors who keep distance, do not want to close to me (physically) or have physical contact with me such as shaking hand. They do not want to get this disease (infection). Some of my friends who know (about his HIV status) also leave me”*.(MP10, single, Belu)

*“I have been working on HIV programs (with a non-governmental organization (NGO) in Yogyakarta) for many years, and I can say that now HIV stigma and discrimination are still occurring but diminishing a lot compared to 5 years ago. I think it is because there have been many HIV programs and activities carried out by community health centers, hospitals and NGOs here. Dissemination of HIV information especially has reached more communities ….”*.(MP18, single, Yogyakarta)

The participants across the study settings commented that social perceptions that associated HIV infection with perceived negative behaviors (e.g., sex with multiple sex partners, engagement in non-marital sexual relations or engagement in sex work) were the driving factors for stigma and discrimination towards PLHIV. The stories of some participants in Yogyakarta showed that such social perceptions seemed to be rooted in their religious thoughts in Islam about extra-marital sex and sex with multiple sex partners as sins and HIV infection as a curse for PLHIV due to their engagement in such sexual behaviors, factors which were not identified in the interviews with participants in Belu. Additionally, social perceptions about HIV as a dangerous, deadly, disgusting and embarrassing infection which had no cure, and the lack of knowledge about HIV which led to the fear of contracting it, were reported as the drivers of discriminatory and stigmatizing attitudes and behaviors towards PLHIV by community members:

*“Social perceptions about HIV are very negative, a disease (infection) of people with negative behaviors, such as women who are sex workers, have multiple sex partners or non-marital sex. …. They perceive HIV as a disgrace for family. Such perceptions influence how other people look at or react towards HIV-positive people …. To be honest, I feel uncomfortable with these perceptions”*.(FP12, married, Yogyakarta)

*“It seems that people in the mosque do not like me (due to his HIV status). During the Friday prayer I sit in the middle and the ones who come after me will shake hand with others but not with me. They are scared of getting HIV. I know that they do not know much about HIV. What they know is that HIV is deadly and there is cure for it. This kind of information spreads in the community and influences the way people (community members) see or interact with me”*.(MP14, divorced, Yogyakarta)

*“People**think that the ones who contract HIV are dirty. They are drug users and female sex workers. HIV is a curse from God to them because their behaviors are not right. That is why many people, including my mom are discriminative towards HIV-positive people like me (her mother was very discriminative to her during the first few months after the HIV diagnosis)”*.(FP24, divorced, Yogyakarta)

*“Many people do not know about HIV and they think that HIV is a disease (infection) of people who have sex with multiple sex partners. They think people get HIV because they often change sex partners, which is something that many people do not accept. That is why HIV positive people like me look bad, negative to their eyes. Perceptions like this spread from mouth to mouth among community members and people are easily influenced by what they hear”*.(MP8, single, Belu)

Some female and male participants across both study settings also described that moral judgements or perceptions about PLHIV as the ones with low moral standing were also drivers of stigma and discrimination towards PLHIV. Their stories indicated that people had such perceptions about PLHIV because they associated HIV with immoral behaviors such as sex with multiple sex partners or engagement in sex work, which also made them disdain and disrespect PLHIV:

*“People have in their heads that HIV transmits because of free sex or sex work, hence many do not respect HIV-positive people. They think we (PLHIV) are immoral because we engage in those immoral behaviors. I can feel it if someone who knows about my (HIV) status and disdains or disrespects me”*.(FP4, single, Yogyakarta)

*“People associate it (HIV) with bad or immoral behaviors. Such perception is common, and it makes people think that all HIV-positive people have immoral behaviors. That is why many people disdain HIV-positive people like me”*.(FP7, remarried, Belu)

*“Many people still look at people living with HIV as the ones who have low moral standing compared to the others (HIV negative people)”*.(MP11, married, Yogyakarta)

### 3.4. Stigma and Discrimination within Healthcare Settings

Several participants, both women (*n* = 13) and men (*n* = 14), in Yogyakarta and Belu reported having experienced some similar discriminatory and stigmatizing attitudes and behaviors of healthcare professionals within healthcare settings where they accessed non-HIV-related healthcare services. For example, the participants described that they received negative labelling or cynical questions from healthcare professionals, experienced a delay of services or were not served due to their HIV status. They also reported that healthcare professionals spread information about their HIV status to others, showed disgusted feeling towards them and were scared of contracting HIV once their HIV status was known:

*“I underwent medical check-up, the laboratory staff (a healthcare professional) asked me: ‘how did you get it (HIV)?’ I got it from my (late) husband, I said. ‘Is your husband dead?’ Yes, I replied. ‘Did your husband like ‘jajan’ (have sex with female sex workers. It literally means eating snack)?’ In their mind, people who contracted HIV must be naughty (sex worker or have sex with multiple sex partners). I have got the same questions before: ‘are you ‘naughty’?’ ‘Do you like jajan?’”*.(FP3, remarried, Yogyakarta)

*“There were nurses who gossiped about my HIV status. They were scared to get close to me or touched me…. There was a nurse who told people within the community that I am sick because of this (HIV). She spread information (about his HIV status) within our community that I get HIV”*.(FP21, widowed, Belu)

*“I experienced discrimination in a healthcare facility, but it was in another ward (dental ward), not in HIV ward (clinic). At that time, I wanted to check my teeth. …. I was honest, I told the dentist that I am HIV-positive. The dentist was shocked and nervous, perhaps she never had patients with HIV. She said to me: ‘please give me a moment, I will talk to my boss (head of the ward)’. She came back and said: ‘I cannot take the decision (to serve him) because we have to have a meeting first.’ I was told to come back in four days. After four days, I came back and received the same treatment. I was not served, and she said the decision has not been made. ….”*.(MP10, separated, Yogyakarta)

A few male participants in both study settings (*n* = 5) also reported that they were verbally discriminated by healthcare professionals in non-HIV wards or clinics. Their HIV status was loudly mentioned in front of other patients and healthcare professionals told each other in front of them to use disposal gloves due to their HIV-positive status, were some instances of verbal discrimination by healthcare professionals towards these participants:

*“I had experienced discrimination. I went to a healthcare facility where I was registered (as indicated in his health insurance) to ask for referral letter (in order to be able to access HIV healthcare services in other healthcare facilities where the services are available), the nurse called out my HIV status. She mentioned it clearly which made other patients surprised. So, it was like my status was open to other people”*.(MP11, married, Yogyakarta)

*“Once my child was admitted to hospital, the nurses told each other in front me to use disposal gloves and said: ‘this kid’s dad has HIV’. What they said made me feel very sad and angry at the same time, but I could not do anything”*.(MP9, married, Belu)

As the consequences of HIV stigma and discrimination by healthcare professionals, some participants across the study settings felt disappointed with healthcare professionals, traumatized, and reluctant to access healthcare services at the same healthcare facilities where they experienced stigma and discrimination. Some decided to hide their HIV status from healthcare professionals, access healthcare services in other healthcare facilities or only consult certain healthcare professionals who they expected to have good treatments or non-discriminatory and stigmatizing attitudes and behaviors. The following comments presented the participants’ perceptions about these aspects:

*“I need lots of courage just to come to a hospital. I was traumatized with hospitals, and my body gets cold if I see hospitals. I was treated very badly by the doctor in a previous hospital. My child was not provided with ARV (antiretroviral) medicines with the reason that there should be a healthy (HIV-negative) family member who accompanied her, otherwise ARV would not be provided. Once a healthy family member of mine accompanied her (to access ARV medicines), the doctor said: ‘wait until her dad is fully recovered’. My husband was sick (HIV-positive and hospitalized). My child who is HIV-positive was not allowed to pup (use the toilet) in the hospital”*.(FP2, widowed, Yogyakarta)

*“After I was rejected (he was not served for tooth extraction in a dental clinic due to his HIV status), I decided not to tell my (HIV) status to any healthcare professionals every time I access (non-HIV-related) healthcare services in any healthcare facilities other than HIV clinic and I do not want to go to that (dental) clinic anymore”*.(MP10, separated, Yogyakarta)

*“I do not want to see her face anymore (a nurse who spread his HIV status to other community members). I do not want to go the community health center either because I might meet her there. So, this (HIV clinic which is a part of a public hospital) is the only place I access healthcare services”*.(MP10, single, Belu)

However, most participants in Yogyakarta acknowledged that HIV stigma and discrimination by healthcare professionals had reduced due to improved healthcare services for HIV patients compared to previous years. They described that more community health centers and hospitals were prepared to provide HIV-related healthcare services, and more healthcare professionals became aware of HIV and served HIV patients professionally. Meanwhile, several participants in Belu commented that HIV stigma and discrimination by healthcare professionals still occurred in healthcare facilities other than HIV clinic due to the lack of knowledge of HIV:

*“Based on my work experience as a companion of HIV-positive people, I would say that now healthcare services for HIV patients have significantly improved compared to 5 or 10 years ago. I experienced that many healthcare professionals were scared to get close to or touch me (physically) back in the early years of the diagnosis (she was diagnosed with HIV in 2007). But now, many of them are aware of HIV and many healthcare facilities are prepared to provide HIV healthcare services to HIV patients. Now HIV patients can mingle with doctors and nurses just like normal without much gap like before, and I think stigma and discrimination by healthcare professionals have reduced”*.(WP16, divorced, Yogyakarta)

*“Stigma and discrimination against HIV-positive patients by nurses still happen. I just had an experience of an unpleasant treatment by nurses in XX community health center a few months ago once I collected the referral letter to bring here (HIV clinic). They avoided me, were scared to serve me. I think many of them do not have proper knowledge about HIV. ….”*.(MP5, single, Belu)

The interrelation of all identified themes in this study and how different factors that facilitated or drove stigma and discrimination towards PLHIV is presented in the diagram (see Figure 1). The diagram shows that HIV stigma and discrimination were experienced by participants (PLHIV) within families, communities, and healthcare facilities. Lack of knowledge of how HIV is transmitted and fear of contracting HIV, which also led to social influence through dissemination of incorrect information about HIV transmission among family, community members, and healthcare providers, were factors that drove stigma and discrimination towards PLHIV by family and community members and healthcare providers. Negative social perceptions, religious thoughts, and negative moral judgement about HIV, PLHIV, and sexual relations were also drivers of HIV stigma and discrimination towards PLHIV in these settings. Family and community members and healthcare providers in these settings may also influence each other’s behaviors and attitudes as they may share the same knowledge, fear, and social, moral and religious thoughts and perceptions.

## 4. Discussion

The paper presents an analysis of the perceptions and experiences of PLHIV about facilitators or drivers of stigma and discrimination towards them within families, communities and healthcare settings in the contexts of Yogyakarta and Belu, Indonesia. It highlights the important role that family and community members played in the participants’ experience of HIV stigma and discrimination, which is in line with the notion of stigma as a process that is linked to the actions of whole groups of people, not simply individual behaviors [28].

Findings in the current paper suggest that HIV stigma and discrimination were experienced by the participants across the study settings within their own families. These were reflected in a range of discriminatory and stigmatizing attitudes and behaviors of family members, such as separations of personal belongings such as clothes and eating utensils from those of other family members, separation from children, ostracism, avoidance, negative stereotype or labelling, and eviction from home, which are consistent with previous study’s results reported elsewhere [13,27,51]. Supporting the constructs of the HIV stigma framework [3] and the findings of previous studies [13,14,19,20,27,51], the current findings suggest that fear of contracting HIV through physical contacts or interactions, and a lack of knowledge about how HIV is transmitted, were major facilitators of discriminatory and stigmatizing attitudes and behaviors towards participants by their family members. The current findings provide further evidence on the process of stigma and discrimination within families, highlighting that HIV stigma and discrimination towards participants were not merely individual attitudes and behaviors, but also a process of social influence by or among family members through provision of incorrect or misleading information about HIV. The findings also indicate that close family ties, a social condition within communities in the study settings, seemed to facilitate the quick influence and spread of such information among family members, which led to stigma and discrimination towards PLHIV, factor that has not been reported in previous literature of HIV stigma [10,11,12].

In line with the concepts in the HIV stigma framework and the reports of previous studies in other settings [3,17,19,20,21,22,52], the current study confirms that HIV stigma and discrimination also experienced by participants within communities where they lived and worked. These manifested in a range of unfair treatments by other community members, such as refusal of eating food they have touched, avoidance of shaking hands or sitting next to them and keeping a distance from them due to the fear of contracting HIV which seemed to be supported by a lack knowledge of the means of HIV transmission [13,14,25,26,27,52]. This study also provides new evidence that discriminatory and stigmatizing attitudes and behaviors of neighbors and friends or community members towards the participants were influenced by negative social and moral perceptions or judgements about HIV and PLHIV, which seemed to be rooted in their religious thoughts about extra-marital sex and sex with multiple sex partners as sins and HIV infection as a curse for PLHIV. These perceptions and thoughts played an important role as drivers of HIV stigma and discrimination by community members towards PLHIV, which have not been explored in much of previous literature on HIV-related stigma [8,11,28]. Such influence seemed to be facilitated by the strong community ties or communal characteristics of communities in the study settings or Indonesia, where people tend to gather and share through family and social activities or events [34,53]. Therefore, HIV stigma and discrimination towards participants in this study reflect not simply individual attitudes and behaviors of each community member, but a process linked to the actions of the whole group of people within communities or societies where the participants lived and worked. The findings support the concept asserting that in societies that emphasize collectivism, such as Indonesia, sociocultural thoughts, norms and values have strong influence on attitudes and behaviors of community members [32,33,34]. In this context, such strong community or social and moral values impacted negatively on participants through discriminatory and stigmatizing attitudes and behaviors of other (non-infected) people.

In line with the previous findings [14,18,21,23,24,51], the current findings suggest that HIV stigma and discrimination also experienced by participants within healthcare facilities where they accessed non-HIV healthcare services. These were reflected in a range of negative attitudes and behaviors of healthcare professionals, such as asking cynical questions to HIV patients, giving negative labelling, spreading information about the HIV status of HIV patients, feeling disgusted towards HIV patients and refusal of care and treatment for HIV patients [14,18,21,23,24,51]. In addition to the fear of contracting HIV, it is plausible to argue that negative social perceptions and moral judgements about HIV and PLHIV could have also been the drivers of such negative attitudes and behaviors of healthcare professionals as they were highly likely to share the same perceptions within communities in the study settings. The current findings also suggest that those HIV stigma and discrimination by healthcare professionals caused traumatized and disappointed feelings for PLHIV, concealment of HIV status, and hindered their access to healthcare services, which are consistent with the previous findings reported elsewhere and HIV stigma concepts applied in this study [3,8,25,27,54,55]. The findings also indicate that health service procedures and policies in healthcare institutions in the study settings, which did not address the needs of HIV patients, constrained the opportunities of PLHIV to access to healthcare services and influenced their health and wellbeing, which are in line with the concept of structural stigma [40,41].

### 4.1. Reflexivity of the Researchers

The role of researchers in the process of knowledge generation is an important aspect that needs to be acknowledged to account for biases and personal interpretations or experiences the researchers bring into a research, to create a balance between their personal understanding and participants’ views, and to improve the trustworthiness of the findings [56]. The researchers in this study have strong educational background in public health and qualitive methods, and many years qualitative research experience in a range of public health issues, including HIV, healthcare services, etc. It is acknowledged that educational background and research experience researchers bring into a research can affect or contribute to the topic under investigation and the interpretations of the research findings [57]. Given the strong educational background and research experience of the researchers in the current study, it is believed that the research questions drove the methodology and methods employed to answer the research questions. For example, prior to designing this study and given the context of the study settings which were in Indonesia where HIV stigma and discrimination towards PLHIV are still common within families, communities, and healthcare settings, the researchers were aware that both women and men living with HIV are highly vulnerable individuals which are difficult to reach. Therefore, the researchers were aware that finding the right channels (e.g., HIV clinics providing HIV services and supports for PLHIV) would be very helpful to disseminate the study information packs to some initial potential participants, who would help disseminate the information further to their eligible friends and colleagues. A qualitative design has been acknowledged to help obtain rich and in-depth personal information or narratives of participants. However, the researchers were aware that the snowball sampling technique used for the recruitment of the participants might be a limitation, but as approaching PLHIV personally would be unethical and was allowed by the ethics committees that approved this study, the snowball sampling technique was considered appropriate for the recruitment of the participants. As is the case for many qualitative studies, the current study’s findings cannot be generalized to all PLHIV, but provide rich and detailed information that can be used to inform HIV-related policies and develop evidence-based programs and interventions to address the needs of PLHIV and stigma and discrimination towards them in Indonesia and other similar settings.

### 4.2. Limitations and Strengths of the Study

The paper cannot be complete without pointing out its potential limitations. The use of snowball sampling technique for the recruitment of the participants and the dissemination of the study information packs through HIV clinics might have led to the recruitment of participants from the same networks, as the participants would have provided information about the study to only those within their networks or who accessed HIV treatment at the clinics. It is, therefore, possible that the study could have been under sampled by potentially not including PLHIV who were outside of the social networks of the current participants. This might have led to incomplete overview of experiences and perceptions about the drivers of HIV stigma and discrimination towards PLHIV. Indeed, all participants were taking ART at the time of the study (by virtue of us recruiting from HIV clinics), and PLHIV who are not taking ART may have different experiences of stigma and discrimination which may have culminated in them disengaging from ART. Further studies may be undertaken specifically with PLHIV who have disengaged from HIV services, and compare their findings to those presented in this paper. However, the strengths of the study were that the purpose of the study was clearly identified, and the use of qualitative design enabled the researcher to explore in-depth the perceptions and experiences of the participants about HIV stigma and discrimination facing them. The use of a framework analysis to guide this qualitative data analysis was also a strength as it ensured the management of qualitative data in a coherent and structured way, and enhanced transparency, rigor, and validity of the analytic process. Moreover, to our knowledge, this is the first qualitative inquiry to focus on HIV stigma as a social process within social context in Indonesia.

## 5. Conclusions

The current paper reports HIV stigma and discrimination towards PLHIV by family, community members and healthcare professionals as more of a process that occurs within social context in Yogyakarta and Belu, Indonesia. It shows that discriminatory and stigmatizing attitudes and behaviors of family, community members and healthcare professionals towards PLHIV were influenced by other family or community members, and facilitated or driven by social perceptions, moral judgement, and religious thoughts within communities or societies where PLHIV lived and interacted. Lack of knowledge about HIV transmission and the fear of contracting HIV through physical or social contacts were identified as drivers of stigma and discrimination toward PLHIV within families, communities, and in healthcare settings. The findings indicate that to respond to HIV stigma and discrimination effectively, targeted interventions are needed, such as specific HIV education at individual, family, healthcare, and societal levels. HIV education for family, community members, and healthcare providers can enhance their knowledge and awareness of HIV and improve acceptance of PLHIV within families, communities, and healthcare settings. Future studies that explore what can be done by government and non-government institutions at policies and practical levels to address HIV stigma and discrimination and improve health service delivery to PLHIV are also recommended.

## Figures and Tables

**Figure 1 ijerph-18-05424-f001:**
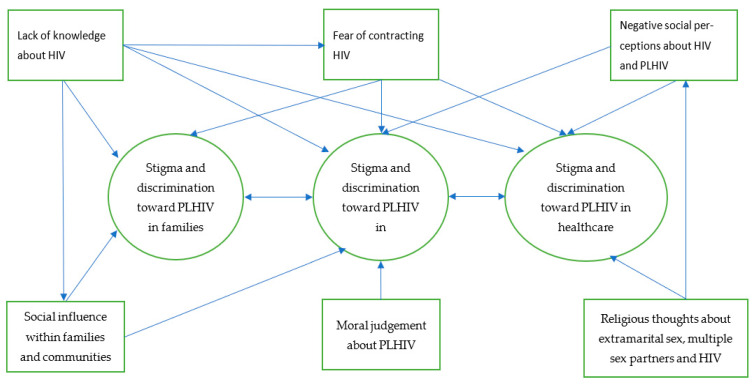
Facilitators or drivers of stigma and discrimination towards PLHIV.

## Data Availability

The data presented in this study are available on request from the corresponding author. The data are not publicly available due to restrictions set by the human research ethics committee.

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
