# Peer review of "Stigma and Discrimination towards People Living with HIV in the Context of Families, Communities, and Healthcare Settings: A Qualitative Study in Indonesia"

_ijerph, 2021, doi:10.3390/ijerph18105424_

Round 1
Reviewer 1 Report
The material is interesting and the topic is relevant. The method seems to have been followed faithfully and the authors were well-positioned to conduct the analysis. Despite these positives, in my view, the paper needs more work before it could be published and I have made some specific suggestions below.
- The literature addressed is not described accurately so far as I can see. Relevant literature should be presented more deeply in order to support the research problem. Further, there is no clear distinction between manuscript sections in terms of the content they report. First, I suggest dividing the section "Introduction" into three components, respectively introduction (explain the general argument of the paper, without going into specific details) background (situate the study concepts within the context of extant knowledge, discuss the international relevance of the concepts) and purpose, creating greater clarity in the analysis of the reader. What is the study's biggest contribution? The contribution should be clearly stated in the introduction.
Data collection
- There are no sources to help the reader understand the qualitative approach and the research paradigm taken in the paper. This needs to be expanded, clarified, and supported by in-text citations.
- When the data were collected? How the research was explained to respondents/participants?
- Interviews: How were the questions chosen? What was the process? The interview guide was developed based on instruments previously used in other studies?
Data analysis
- The process of analysis should be made as transparent as possible (notwithstanding the conceptual and theoretical creativity that typically characterises qualitative research).
- Did participants provide feedback on the findings?
- Were emotions and notes added during transcription to reflect the meaning in the words or did you trust your memory?
- Further information should be given of how authors had adopted open attitudes. Did they use reflexive bracketing to achieve openness? The researcher’s own position should clearly be stated. For example, how they examined their own role, possible bias, and influence on the research (reflexivity)? What experience or training did the researchers have?
Findings
- Findings were not explicitly interrelated; there was no presentation of a structure. I suggest an illustration/diagram for a better analysis.
Discussion
- There are some conclusions drawn which have neither literature review nor research to support them. It is also not entirely clear whether this is deductive or inductive logic employed here.
- The recommendations/implications for practice/research/education/management should have been approached in greater depth. Please discuss whether or how the findings can be transferred to other populations.
EDITORIAL COMMENTS
- The manuscript will serve a broad audience of students, researchers, and practitioners, however, the manuscript needs to be carefully and attentively proofread, because some sentences are awkwardly constructed, punctuation is deficient, and therefore reading is occasionally difficult to follow. The English of this manuscript should be reviewed by a native-English speaker.
- Many of the cited references are somewhat dated.
Author Response
A point-by-point response
Reviewer 1:
The material is interesting and the topic is relevant. The method seems to have been followed faithfully and the authors were well-positioned to conduct the analysis. Despite these positives, in my view, the paper needs more work before it could be published and I have made some specific suggestions below.
- The literature addressed is not described accurately so far as I can see. Relevant literature should be presented more deeply in order to support the research problem. Further, there is no clear distinction between manuscript sections in terms of the content they report. First, I suggest dividing the section "Introduction" into three components, respectively introduction (explain the general argument of the paper, without going into specific details) background (situate the study concepts within the context of extant knowledge, discuss the international relevance of the concepts) and purpose, creating greater clarity in the analysis of the reader. What is the study's biggest contribution? The contribution should be clearly stated in the introduction.
Response:
- We have improved the use and description of literature in the introduction section.
- Introduction section has been divided into three components as suggested
- Introduction (explain the general argument of the paper, without going into specific details):
Human immunodeficiency virus and acquired immune deficiency syndrome (HIV/AIDS) have been reported as a major public health problem for decades, with an estimated 38 million people globally living with the infection (1, 2). It is also well acknowledged that a diagnosis of HIV infection causes a range of detrimental impacts for people living with HIV (PLHIV) (3-6). Stigma which is often manifested as discrimination or unfair treatment by other (HIV negative) people is one of the major negative impacts on PLHIV in many settings (3, 7, 8). Despite positive achievements in the response to the epidemic globally, increased universal access to antiretroviral therapy (ART) and reduction of infection across the world, HIV stigma and discrimination are still a global problem (2, 9).
- Background (situate the study concepts within the context of extant knowledge, discuss the international relevance of the concepts)
Previous studies and reports have suggested that HIV stigma and discrimination towards PLHIV occur within families, communities and in healthcare settings (10-12). Several studies have reported that stigma and discrimination towards PLHIV often occur within families by parents, siblings, relatives or in-laws (13, 14). These are reflected in a range of discriminatory treatment and behaviours, including refusal by others to share food and rooms with PLHIV, separation of personal belongings and eating utensils of PLHIV from those of other family members, isolation of PLHIV by their own family, including exclusion from usual family activities such as cooking and family gathering (13-20). HIV stigma and discrimination towards PLHIV have also been reported to be inflicted by neighbours, friends and co-workers and these often manifest as rejection, neglect, avoidance, ridicule, verbal abuse, insult and harassment (14-17, 19, 21, 22). Similar acts of stigma and discrimination towards PLHIV have also reported within healthcare facilities or settings by healthcare professionals in a variety of ways, including criticising, blaming, shouting at or throwing health records on patients’ faces, and neglecting or refusal of care and treatment, and through unnecessary referral to other healthcare facilities (14, 18, 21, 23, 24). The fear of contracting HIV through physical, social and healthcare-related contacts and interactions, and the lack of knowledge about how HIV is transmitted, have been reported as the main drivers of HIV stigma and discrimination in these settings (13, 14, 25-27). HIV stigma and discrimination have also been reported to bring negative impacts on psychological state, health outcomes and social life of PLHIV. They are associated with stress, anxiety, depression and low quality of life for PLHIV (3-6). HIV stigma and discrimination have been reported to negatively influence access and adherence to ART or HIV prevention and treatment efforts and disrupt social relationships of PLHIV with their families, relatives, friends and neighbours (3-6).
- Purpose, creating greater clarity in the analysis of the reader. What is the study's biggest contribution? The contribution should be clearly stated in the introduction.
Although many other studies have explored HIV stigma and discrimination towards PLHIV in different settings, the majority of these studies have focused on stigma at individual level, including studying the attitudes and behaviours of HIV non-infected individuals towards PLHIV (8, 11, 28), leaving the gap in knowledge about how HIV stigma and discrimination are enacted as a social process. Therefore, the aim of this study was to explore HIV stigma and discrimination beyond individuals and to assess how they are enacted as a social process in the context of families, communities and healthcare settings. The overall aim is to contribute to the understanding of drivers of stigma that arise within social contexts in Indonesia where the influence of strong family and community values, norms, ties, and religious thoughts on stigma and discrimination towards PLHIV have not been addressed in previous studies (8, 11, 28)]. As HIV stigma and discrimination are reported to occur across settings in Indonesia (14, 27, 29-31) and Indonesian society is influenced strongly by family and community values, norms, ties, influences, and religious (32-34), it is important to unpack this complex societal structure, to further inform how social processes influence and propagate discriminatory and stigmatising attitudes and behaviours towards PLHIV. This information is crucial as will provide significant contribution to the current body of knowledge on the topic and inform policies and practices within government and non-governmental institutions and organisations to address social impacts of HIV and improve health outcomes of PLHIV in Indonesia and globally.
Data collection
- There are no sources to help the reader understand the qualitative approach and the research paradigm taken in the paper. This needs to be expanded, clarified, and supported by in-text citations.
Response:
- Line 130-138: This paper presents data from a large-scale qualitative study exploring the views or perceptions of women and men living with HIV about HIV risk factors and impacts and their access to HIV healthcare services in Yogyakarta and Belu, Indonesia. The qualitative design was used as it has been found appropriate and effective when exploring participants’ perspectives and deep insight of their real-life experiences (42, 43). It facilitated the exploration of the participants’ stories, understandings and interpretations about the supporting factors for HIV transmission among them, and drivers of stigma and discrimination against them by other (non-infected) people (44-46)]. It also enabled the researchers to explore and understand values and meanings the participants had in relation to HIV stigma and discrimination facing them in their daily lives (42, 47).
- When the data were collected? How the research was explained to respondents/participants?
Response:
- Line 248-250: Data collection was conducted from June to November 2019 using one-on-one in-depth interviews in a rented house close to the HIV clinic in Yogyakarta and a private room at the HIV clinic in Belu.
- Line 224-237: They were advised about the purpose of the study and the voluntary nature of their participation prior to the interviews through the study information packs distributed to during the recruitment process and again in person by the field researcher prior to each interview. They were informed about their right to withdraw from the study at any time, without consequence, if they felt uncomfortable with the questions being asked during the interviews. They were also advised about the approximate interview duration of 45 to 90 min and that audio recording and notes would be undertaken during the interviews. The participants were assured about confidentiality and anonymity of the data or information they provided in the interview, which was maintained by assigning a specific study identification letter and number for each participant (e.g., FP1, FP2, …., or MP1, MP2, …. FP stands for Female Participant and MP stands for Male Participant). Reimbursement of IDR 100,000 (±USD 7) for transport and time of the participants was provided for each of them after the interview. The informed consent form was provided for each participant, signed, and returned to the researcher prior to the interview.
- Interviews: How were the questions chosen? What was the process? The interview guide was developed based on instruments previously used in other studies?
Response:
- Line 153-164: Regarding HIV stigma and discrimination, the interview topics explored participants’ perceptions and experiences of HIV stigma and discrimination. The researcher probed further about attitudes and behaviours of family and community members and healthcare providers towards them. Moreover, the impacts of unfair treatments and attitudes of other people towards them were explored. Additionally, participants were asked about perspectives regarding drivers or facilitators of and mechanisms or processes through which those facilitators or drivers contributed to stigma and discrimination. Participants’ perspectives about how social influence among family and community members which led to stigma and discrimination against them and other PLHIV were also explored. The development of interview questions was informed by the findings of previous studies and the theoretical framework used in this study.
Data analysis
- The process of analysis should be made as transparent as possible (notwithstanding the conceptual and theoretical creativity that typically characterises qualitative research).
Response:
- Line 176-217: The digital recordings of the interviews were manually transcribed verbatim in Bahasa by the first author (NKF). The transcripts were then imported to NVivo 12 where the comprehensive data analysis was performed, which was guided by a framework analysis for qualitative data by Ritchie and Spencer (48)]. The framework was used as it helped the management of qualitative data in a coherent and structured way, and guided the analytic process in a rigorous, transparent and valid way. This framework suggests five steps of qualitative data analysis, that are: (i) Familiarisation with the data or transcripts, which was done by repeatedly reading each transcript, breaking down the data into small chunks of data, and making comments or labels to the data extracts of each individual transcript. The transcription of the audio recordings, which was manually performed using a laptop, had been started by the field researcher following each interview during the data collection process. At this stage, emotions or notes undertaken during interviews were added into each individual transcript. Thus, the process of familiarisation with the data had been started alongside the data collection process; (ii) identification of a thematic framework by writing down key issues and concepts that recurrently emerged from the data, which was performed after importing each individual transcript with the initial comments, codes, labels into NVivo. Themes that were used to form the thematic framework were derived from both the HIV stigma framework used in this study and the collected data. The identification of the thematic framework was an iterative process that involved changing and refining themes; (iii) indexing the data which was comprehensively performed using NVivo. The process of indexing (coding) was started by making open codes to data extracts of each individual transcript resulting in a long list of open codes or nodes. This was followed by close coding to identify similar or redundant nodes or codes and reduce the long list of open codes to a manageable number, and then, nodes or codes that seemed to fall into the same themes and sub-themes were grouped together; (iv) charting data through arranging appropriate thematic references in a summary of chart which enabled comparison across interviews and within each interview. This was performed by reorganising and summarising codes from each individual interview transcript, that had been grouped into separate themes in the previous section, and putting them together under each theme; and (v) mapping and interpretation of the data through which data were examined and interpreted (48, 49). Based on the pieces of data that had been indexed and charted in the previous steps, the researchers systematically pulled together key characteristics of the data, mapped and interpreted data set as a whole. Transcription, coding and analysis were conducted in Bahasa, and quotations for publication purposes were translated into English by NKF. To maintain the accuracy of the translation and credibility of the findings, checking and rechecking transcripts against the translated interpretations or examination of meaning in both source (Bahasa) and target (English) languages were done during the analysis (50)]. Analysis was primarily undertaken by NKF, although team-based analysis was undertaken at regular research team meetings whereby all authors undertook independent analysis and then team decisions were made about the validity of the final themes and interpretation.
- Did participants provide feedback on the findings?
Response:
- Line 169-173: We did not offer an opportunity for participants to read and correct the information provided after the transcription due to the sensitivity of the topic and to prevent the possibility of the transcripts being received and read by their family members, which might divulge the participants’ HIV status, in case they had not disclosed it to family members.
- Were emotions and notes added during transcription to reflect the meaning in the words or did you trust your memory?
Response:
- Line 184-188: The transcription of the audio recordings, which was manually performed using a laptop, had been started by the field researcher following each interview during the data collection process. At this stage, emotions or notes undertaken during interviews were added into each individual transcript.
- Further information should be given of how authors had adopted open attitudes. Did they use reflexive bracketing to achieve openness? The researcher’s own position should clearly be stated. For example, how they examined their own role, possible bias, and influence on the research (reflexivity)? What experience or training did the researchers have?
Response:
- Line 638-669
4.1 Reflexivity of the researchers
The role of researchers in the process of knowledge generation is an important aspect that needs to be acknowledged to account for biases and personal interpretations or experiences the researchers bring into a research, to create a balance between their personal understanding and participants’ views, and to improve the trustworthiness of the findings (57). The researchers in this study have strong educational background in public health and qualitive methods, and many years qualitative research experience in a range of public health issues, including HIV, healthcare services, etc. It is acknowledged that educational background and research experience researchers bring into a research can affect or contribute to the topic under investigation and the interpretations of the research findings (58). Given the strong educational background and research experience of the researchers in the current study, it is believed that the research questions drove the methodology and methods employed to answer the research questions. For example, prior to designing this study and given the context of the study settings which were in Indonesia where HIV stigma and discrimination towards PLHIV are still common within families, communities and healthcare settings, the researchers were aware that both women and men living with HIV are highly vulnerable individuals which are difficult to reach. Therefore, the researchers were aware that finding the right channels (e.g., HIV clinics providing HIV services and supports for PLHIV) would be very helpful to disseminate the study information packs to some initial potential participants, who would help disseminate the information further to their eligible friends and colleagues. A qualitative design has been acknowledged to help obtain rich and in-depth personal information or narratives of participants. However, the researchers were aware that the snowball sampling technique used for the recruitment of the participants might be a limitation, but as approaching PLHIV personally would be unethical and was allowed by the ethics committees that approved this study, the snowball sampling technique was considered appropriate for the recruitment of the participants. As is the case for many qualitative studies, the current study’s findings cannot be generalised to all PLHIV, but provide rich and detailed information that can be used to inform HIV-related policies and develop evidence-based programs and interventions to address the needs of PLHIV and stigma and discrimination towards them in Indonesia and other similar settings.
Findings
- Findings were not explicitly interrelated; there was no presentation of a structure. I suggest an illustration/diagram for a better analysis.
Response:
- Line 527-559: A diagram has been provided.
Discussion
- There are some conclusions drawn which have neither literature review nor research to support them. It is also not entirely clear whether this is deductive or inductive logic employed here.
Response:
- The discussion section has been improved.
- The recommendations/implications for practice/research/education/management should have been approached in greater depth. Please discuss whether or how the findings can be transferred to other populations.
Response:
- Line 703-708: The findings indicate that in order to respond to HIV stigma and discrimination effectively, targeted interventions are needed, such as specific HIV education at individual, family, healthcare and societal levels. HIV education for family, community members and healthcare providers can enhance their knowledge and awareness of HIV and improve acceptance of PLHIV within families, communities and healthcare settings.
- Line 664-668: As is the case for many qualitative studies, the current study’s findings cannot be generalised to all PLHIV, but provide rich and detailed information that can be used to inform HIV-related policies and develop evidence-based programs and interventions to address the needs of PLHIV and stigma and discrimination towards them in Indonesia and other similar settings.
Reviewer 2 Report
This study using a qualitative survey to explore the drivers and processes of HIV stigma and discrimination from the perspective of PLHIV in the setting of family, community, and healthcare providers in Indonesia. The study is generally well written and organized which has substantially discerned the impact of this study on the field; However, a number of limitations are noted that reduce the ability to discern the impact:
Intro:
I would recommend expanding the impact of stigma on health care outcomes, social life, and behaviors rather than list the specific stigma and discrimination behaviors. This would give a reader the potential benefits of reducing stigma and discrimination.
It may be true that strong family and community values influence the stigma in Indonesia. Authors may provide more rationales why this study is necessary for Indonesia (e.g. multi-religion).
Method:
Please provide the number of people interviewed in Yogyakarta and Belu, respectively.
Discussion:
The same phrase “constructs of the HIV stigma framework and findings of previous studies” were used multiple times, please consider rephrase them.
Authors could discuss the shared reasons for stigma across three settings. For example, lacking knowledge of HIV transmission caused the separation in family, community, and healthcare providers.
Authors could discuss more the implications of informed policymaking.
Author Response
Reviewer 2:
This study using a qualitative survey to explore the drivers and processes of HIV stigma and discrimination from the perspective of PLHIV in the setting of family, community, and healthcare providers in Indonesia. The study is generally well written and organized which has substantially discerned the impact of this study on the field; However, a number of limitations are noted that reduce the ability to discern the impact:
Intro:
I would recommend expanding the impact of stigma on health care outcomes, social life, and behaviors rather than list the specific stigma and discrimination behaviors. This would give a reader the potential benefits of reducing stigma and discrimination.
Response:
- Line 72-77: HIV stigma and discrimination have also been reported to bring negative impacts on psychological state, health outcomes and social life of PLHIV. They are associated with stress, anxiety, depression and low quality of life for PLHIV (3-6). They have been reported to negatively influence access and adherence to ART or HIV prevention and treatment efforts and disrupt social relationships of PLHIV with their families, relatives, friends and neighbours (3-6).
It may be true that strong family and community values influence the stigma in Indonesia. Authors may provide more rationales why this study is necessary for Indonesia (e.g. multi-religion).
Response:
- Line 83-95: Therefore, the aim of this study was to explore HIV stigma and discrimination beyond individuals and to assess how they are enacted as a social process in the context of families, communities and healthcare settings. The overall aim is to contribute to the understanding of drivers of stigma that arise within social contexts in Indonesia where the influence of strong family and community values, norms, ties, and religious thoughts on stigma and discrimination towards PLHIV have not been addressed in previous studies (8, 11, 28)]. As HIV stigma and discrimination are reported to occur across settings in Indonesia (14, 27, 29-31) and Indonesian society is influenced strongly by family and community values, norms, ties, influences, and religious (32-34), it is important to unpack this complex societal structure, to further inform how social processes influence and propagate discriminatory and stigmatising attitudes and behaviours towards PLHIV.
Method:
Please provide the number of people interviewed in Yogyakarta and Belu, respectively.
Response:
- Line 145-146: The recruitment process took three months, with 92 PLHIV (52 women and 40 men; 46 in Yogyakarta and 46 in Belu) participating in the study.
Discussion:
The same phrase “constructs of the HIV stigma framework and findings of previous studies” were used multiple times, please consider rephrase them.
Response:
- This phrase has been improved throughout the discussion section.
Authors could discuss the shared reasons for stigma across three settings. For example, lacking knowledge of HIV transmission caused the separation in family, community, and healthcare providers.
Response:
- Line 567-579: Findings in the current paper suggest that HIV stigma and discrimination were experienced by the participants across the study settings occurred within their own families. These were reflected in a range of discriminatory and stigmatising attitudes and behaviours of family members, such as separations of personal belongings like clothes and eating utensils from those of other family members, separation from children, ostracism, avoidance, negative stereotype or labelling, and eviction from home, which are consistent with previous study’s results reported elsewhere (13, 27, 51). Supporting the constructs of the HIV stigma framework (3)] and the findings of previous studies (13, 14, 19, 20, 27, 51), the current findings suggest that fear of contracting HIV through physical contacts or interactions, and a lack of knowledge about the means of HIV transmission, were major facilitators of discriminatory and stigmatising attitudes and behaviours towards participants by their family members.
- Line 588-595: In line with the concepts in the HIV stigma framework and the reports of previous studies in other settings (3, 19-22, 52, 53)], the current study confirms that HIV stigma and discrimination also experienced by participants within communities where they lived and worked. These manifested in a range of unfair treatments by other community members, such as refusal of eating food they have touched, avoidance of shaking hands or sitting next to them and keeping a distance from them due to the fear of contracting HIV which seemed to be supported by a lack knowledge of the means of HIV transmission (13, 14, 25-27, 53). Authors could discuss more the implications of informed policy making.
Authors could discuss more the implications of informed policy making.
Response:
- Line 663-667: As is the case for many qualitative studies, the current study’s findings cannot be generalised to all PLHIV, but provide rich and detailed information that can be used to inform HIV-related policies and develop evidence-based programs and interventions to address the needs of PLHIV and stigma and discrimination towards them in Indonesia and other similar settings.
Round 2
Reviewer 1 Report
After reviewing the above-mentioned manuscript, now reformulated as previously suggested to the authors, it is possible to ascertain a significant improvement in the consolidation of theoretical and methodological aspects, which I am concerned about, decisive to the study quality.